# Do Adversarially Robust ImageNet Models Transfer Better?

**Hadi Salman**[*]
hadi.salman@microsoft.com
Microsoft Research

**Andrew Ilyas**[*]
ailyas@mit.edu
MIT

**Logan Engstrom**
engstrom@mit.edu
MIT

**Ashish Kapoor**
akapoor@microsoft.com
Microsoft Research

**Aleksander Mądry**
madry@mit.edu
MIT

## Abstract

Transfer learning is a widely-used paradigm in which models pre-trained on standard datasets can efficiently adapt to downstream tasks. Typically, better pre-trained models yield better transfer results, suggesting that initial accuracy is a key aspect of transfer learning performance. In this work, we identify another such aspect: we find that adversarially robust models, while less accurate, often perform better than their standard-trained counterparts when used for transfer learning. Specifically, we focus on adversarially robust ImageNet classifiers, and show that they yield improved accuracy on a standard suite of downstream classification tasks. Further analysis uncovers more differences between robust and standard models in the context of transfer learning. Our results are consistent with (and in fact, add to) recent hypotheses stating that robustness leads to improved feature representations. Our code and models are available at https://github.com/Microsoft/robust-models-transfer.

## 1 Introduction

Deep neural networks currently define state-of-the-art performance across many computer vision tasks. When large quantities of labeled data and computing resources are available, models perform well when trained from scratch. However, in many practical settings there is insufficient data or compute for this approach to be viable. In these cases, *transfer learning* [Don+14; Sha+14] has emerged as a simple and efficient way to obtain performant models. Broadly, transfer learning refers to any machine learning algorithm that leverages information from one ("source") task to better solve another ("target") task. A prototypical transfer learning pipeline in computer vision (and the focus of our work) starts with a model trained on the ImageNet-1K dataset [Den+09; Rus+15], and then refines this model for the target task.

Though the exact underpinnings of transfer learning are not fully understood, recent work has identified factors that make pre-trained ImageNet models amenable to transfer learning. For example, [HAE16; Kol+19] investigate the effect of the source dataset; Kornblith, Shlens, and Le [KSL19] find that pre-trained models with higher ImageNet accuracy also tend to transfer better; Azizpour et al. [Azi+15] observe that increasing depth improves transfer more than increasing width.

**Our contributions.** In this work, we identify another factor that affects transfer learning performance: adversarial robustness [Big+13; Sze+14]. We find that despite being less accurate on ImageNet, adversarially robust neural networks match or improve on the transfer performance of their standard counterparts. We first establish this trend in the "fixed-feature" setting, in which one trains

Table 1: Transfer learning performance of robust and standard ImageNet models on 12 downstream classification tasks. For each type of model, we compute maximum accuracy (averaged over three random trials) over training parameters, architecture, and (for robust models) robustness level $\varepsilon$.

| | | Dataset | | | | | | | | | | | |
|---|---|---|---|---|---|---|---|---|---|---|---|---|---|
| Mode | Model | Aircraft | Birdsnap | CIFAR-10 | CIFAR-100 | Caltech-101 | Caltech-256 | Cars | DTD | Flowers | Food | Pets | SUN397 |
| **Fixed-feature** | **Robust** | **44.14** | **50.72** | **95.53** | **81.08** | **92.76** | **85.08** | **50.67** | **70.37** | 91.84 | **69.26** | **92.05** | **58.75** |
| | **Standard** | 38.69 | 48.35 | 81.31 | 60.14 | 90.12 | 82.78 | 44.63 | 70.09 | **91.90** | 65.79 | 91.83 | 55.92 |
| **Full-network** | **Robust** | 86.24 | **76.55** | **98.68** | **89.04** | **95.62** | **87.62** | 91.48 | **76.93** | **97.21** | **89.12** | **94.53** | **64.89** |
| | **Standard** | **86.57** | 75.71 | 97.63 | 85.99 | 94.75 | 86.55 | **91.52** | 75.80 | 97.04 | 88.64 | 94.20 | 63.72 |

a linear classifier on top of features extracted from a pre-trained network. Then, we show that this trend carries forward to the more complex "full-network" transfer setting, in which the pre-trained model is entirely fine-tuned on the relevant downstream task. We carry out our study on a suite of image classification tasks (summarized in Table 1), object detection, and instance segmentation.

Our results are consistent with (and in fact, add to) recent hypotheses suggesting that adversarial robustness leads to improved feature representations [Eng+19a; AL20]. Still, future work is needed to confirm or refute such hypotheses, and more broadly, to understand what properties of pre-trained models are important for transfer learning.

## 2 Motivation: Fixed-Feature Transfer Learning

In one of the most basic variants of transfer learning, one uses the source model as a feature extractor for the target dataset, then trains a simple (often linear) model on the resulting features. In our setting, this corresponds to first passing each image in the target dataset through a pre-trained ImageNet classifier, and then using the outputs from the penultimate layer as the image's feature representation. Prior work has demonstrated that applying this "fixed-feature" transfer learning approach yields accurate classifiers for a variety of vision tasks and often out-performs task-specific handcrafted features [Sha+14]. However, we still do not completely understand the factors driving transfer learning performance.

**How can we improve transfer learning?** Both conventional wisdom and evidence from prior work [Cha+14; SZ15; KSL19; Hua+17] suggests that accuracy on the source dataset is a strong indicator of performance on downstream tasks. In particular, Kornblith, Shlens, and Le [KSL19] find that pre-trained ImageNet models with higher accuracy yield better fixed-feature transfer learning results.

Still, it is unclear if improving ImageNet accuracy is the only way to improve performance. After all, the behaviour of fixed-feature transfer is governed by models' learned representations, which are not fully described by source-dataset accuracy. These representations are, in turn, controlled by the *priors* that we put on them during training. For example, the use of architectural components [UVL17], alternative loss functions [Mur+18], and data augmentation [VM01] have all been found to put distinct priors on the features extracted by classifiers.

**The adversarial robustness prior.** In this work, we turn our attention to another prior: *adversarial robustness*. Adversarial robustness refers to a model's invariance to small (often imperceptible) perturbations of its inputs. Robustness is typically induced at training time by replacing the standard empirical risk minimization objective with a robust optimization objective [Mad+18]:

$$\min_\theta \mathbb{E}_{(x,y)\sim D}\left[\mathcal{L}(x,y;\theta)\right] \implies \min_\theta \mathbb{E}_{(x,y)\sim D}\left[\max_{\|\delta\|_2\leq\varepsilon}\mathcal{L}(x+\delta,y;\theta)\right], \tag{1}$$

where $\varepsilon$ is a hyperparameter governing how invariant the resulting "adversarially robust model" (more briefly, "robust model") should be. In short, this objective asks the model to minimize risk on the training datapoints while also being locally stable in the (radius-$\varepsilon$) neighbourhood around each of these points. (A more detailed primer on adversarial robustness is given in Appendix E.)

Adversarial robustness was originally studied in the context of machine learning security [Big+13; BR18; CW17; Ath+18] as a method for improving models' resilience to adversarial examples [GSS15; Mad+18]. However, a recent line of work has studied adversarially robust models in their own right, casting (1) as a prior on learned feature representations [Eng+19a; Ily+19; Jac+19; ZZ19].

**Should adversarial robustness help fixed-feature transfer?**    It is, a priori, unclear what to expect from an "adversarial robustness prior" in terms of transfer learning. On one hand, robustness to adversarial examples may seem somewhat tangential to transfer performance. In fact, adversarially robust models are known to be significantly less accurate than their standard counterparts [Tsi+19; Su+18; Rag+19; Nak19], suggesting that using adversarially robust feature representations should hurt transfer performance.

On the other hand, recent work has found that the feature representations of robust models carry several advantages over those of standard models. For example, adversarially robust representations typically have better-behaved gradients [Tsi+19; San+19; ZZ19; KCL19] and thus facilitate regularization-free feature visualization [Eng+19a] (cf. Figure 1a). Robust representations are also approximately invertible [Eng+19a], meaning that unlike for standard models [MV15; DB16], an image can be approximately reconstructed directly from its robust representation (cf. Figure 1b). More broadly, Engstrom et al. [Eng+19a] hypothesize that by forcing networks to be invariant to signals that humans are also invariant to, the robust training objective leads to feature representations that are more similar to what humans use. This suggests, in turn, that adversarial robustness might be a desirable prior from the point of view of transfer learning.

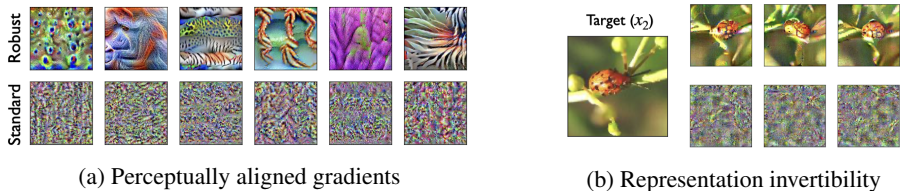

(a) Perceptually aligned gradients                    (b) Representation invertibility

Figure 1: Adversarially robust (top) and standard (bottom) representations: robust representations allow (a) feature visualization without regularization; (b) approximate image inversion by minimizing distance in representation space. Figures reproduced from Engstrom et al. [Eng+19a].

**Experiments.**    To resolve these two conflicting hypotheses, we use a test bed of 12 standard transfer learning datasets (all the datasets considered in [KSL19] as well as Caltech-256 [GHP07]) to evaluate fixed-feature transfer on standard and adversarially robust ImageNet models. We considere four ResNet-based architectures (ResNet-{18,50}, WideResNet-50-x{2,4}), and train models with varying robustness levels $\varepsilon$ for each architecture (for the full experimental setup, see Appendix A).

In Figure 2, we compare the downstream transfer accuracy of a standard model to that of the best robust model with the same architecture (grid searching over $\varepsilon$). The results indicate that robust networks consistently extract better features for transfer learning than standard networks—this effect is most pronounced on Aircraft, CIFAR-10, CIFAR-100, Food, SUN397, and Caltech-101. Due to computational constraints, we could not train WideResNet-50-4x models at the same number of robustness levels $\varepsilon$, so a coarser grid was used. It is thus likely that a finer grid search over $\varepsilon$ would further improve results (we discuss the role of $\varepsilon$ in more detail in Section 4.3).

## 3    Adversarial Robustness and Full-Network Fine Tuning

A more expensive but often better-performing transfer learning method uses the pre-trained model as a weight initialization rather than as a feature extractor. In this "full-network" transfer learning setting, we update all of the weights of the pre-trained model (via gradient descent) to minimize loss on the target task. Kornblith, Shlens, and Le [KSL19] find that for standard models, performance on full-network transfer learning is highly correlated with performance on fixed-feature transfer learning. Therefore, we might hope that the findings of the last section (i.e., that adversarially robust models transfer better) also carry over to this setting. To resolve this conjecture, we consider

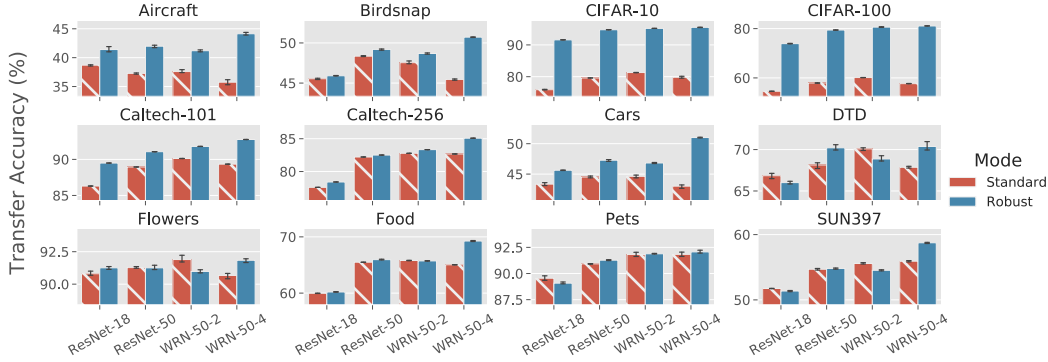

Figure 2: **Fixed-feature** transfer learning results using standard and robust models for the 12 downstream image classification tasks considered. Following [KSL19], we record re-weighted accuracy for the unbalanced datasets, and raw accuracy for the others (cf. Appendix A). Error bars denote the maximum and minimum error attained over three random trials. A similar plot with ten random trials is in Appendix F.

three applications of full-network transfer learning: downstream image classification (i.e., the tasks considered in Section 2), object detection, and instance segmentation.

## 3.1 Downstream image classification

We first recreate the setup of Section 2: we perform full-network transfer learning to adapt the robust and non-robust pre-trained ImageNet models to the same set of 12 downstream classification tasks. The hyperparameters for training were found via grid search (cf. Appendix A). Our findings are shown in Figure 3—just as in fixed-feature transfer learning, robust models match or improve on standard models in terms of transfer learning performance.

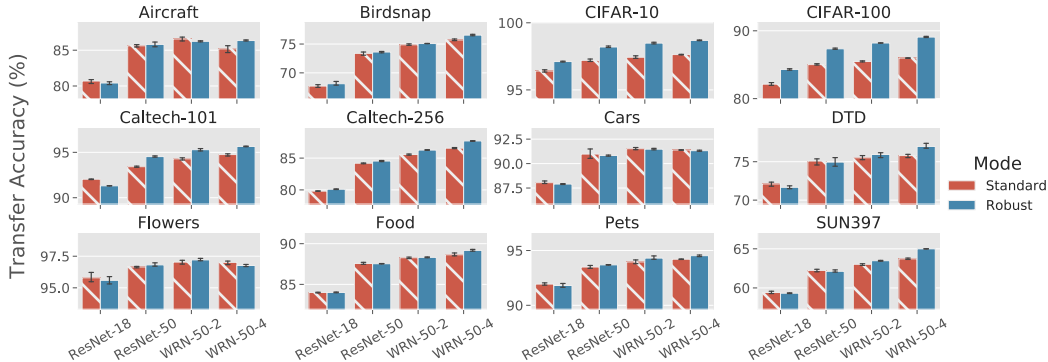

Figure 3: **Full-network** transfer learning results using standard and robust models for the 12 downstream image classification tasks considered. Following [KSL19], we record re-weighted accuracy for the unbalanced datasets, and raw accuracy for the others (cf. Appendix A). Error bars denote the maximum and minimum error attained over three random trials. A similar plot with ten random trials is in Appendix F.

## 3.2 Object detection and instance segmentation

It is standard practice in data-scarce object detection or instance segmentation tasks to initialize earlier model layers with weights from ImageNet-trained classification networks. We study the benefits of using robustly trained networks to initialize object detection and instance segmentation models, and find that adversarially robust networks consistently outperform standard networks.

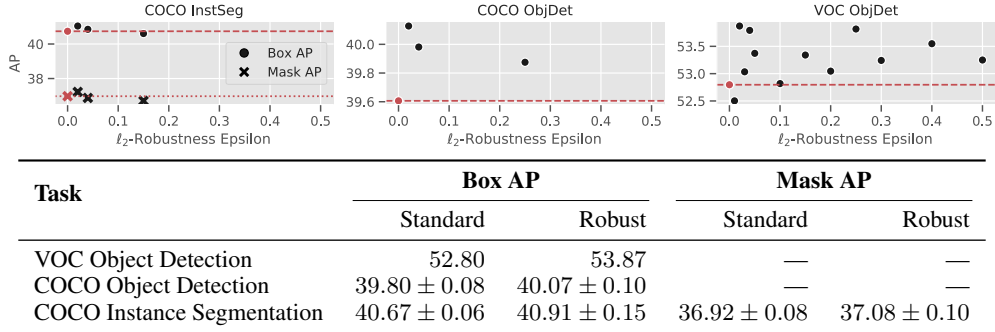

| Task | Box AP | | Mask AP | |
|---|---|---|---|---|
| | Standard | Robust | Standard | Robust |
| VOC Object Detection | 52.80 | 53.87 | — | — |
| COCO Object Detection | $39.80 \pm 0.08$ | $40.07 \pm 0.10$ | — | — |
| COCO Instance Segmentation | $40.67 \pm 0.06$ | $40.91 \pm 0.15$ | $36.92 \pm 0.08$ | $37.08 \pm 0.10$ |

Figure 4: AP of instance segmentation and object detection models with backbones initialized with $\varepsilon$-robust models before training. Robust backbones generally lead to better AP, and the best robust backbone always outperforms the standardly trained backbone for every task. COCO results averaged over four runs due to computational constraints; $\pm$ represents standard deviation.

**Experimental setup.** We evaluate with benchmarks in both object detection (PASCAL Visual Object Classes (VOC) [Eve+10] and Microsoft COCO [Lin+14]) and instance segmentation (Microsoft COCO). We train systems using default models and hyperparameter configurations from the Detectron2 [Wu+19] framework (i.e., we do not perform any additional hyperparameter search). Appendix C describes further experimental details and more results.

We first study object detection. We train Faster R-CNN FPN [Lin+17] models with varying ResNet-50 backbone initializations. For VOC, we initialize with one standard network, and twelve adversarially robust networks with different values of $\varepsilon$. For COCO, we only train with three adversarially robust models (due to computational constraints). For instance segmentation, we train Mask R-CNN FPN models [He+17] while varying ResNet-50 backbone initialization. We train three models using adversarially robust initializations, and one model from a standardly trained ResNet-50. Figure 4 summarizes our findings: the best robust backbone initializations outperform standard models.

## 4 Analysis and Discussion

Our results from the previous section indicate that robust models match or improve on the transfer learning performance of standard ones. In this section, we take a closer look at the similarities and differences in transfer learning between robust networks and standard networks.

### 4.1 ImageNet accuracy and transfer performance

In Section 2, we discussed a potential tension between the desirable properties of robust network representations (which we conjectured would improve transfer performance) and the decreased accuracy of the corresponding models (which, as prior work has established, should hurt transfer performance). We hypothesize that robustness and accuracy have counteracting yet separate effects: that is, higher accuracy improves transfer learning for a fixed level of robustness, and higher robustness improves transfer learning for a fixed level of accuracy.

To test this hypothesis, we first study the relationship between ImageNet accuracy and transfer accuracy for each of the robust models that we trained. Under our hypothesis, we should expect to see a deviation from the direct linear accuracy-transfer relation observed by [KSL19], due to the confounding factor of varying robustness. The results (cf. Figure 5; similar results for full-network transfer in Appendix F) support this. Indeed, we find that the previously observed linear relationship between accuracy and transfer performance is often violated once robustness aspect comes into play.

In even more direct support of our hypothesis (i.e., that robustness and ImageNet accuracy have opposing yet separate effects on transfer), we find that when the robustness level is held fixed, the accuracy-transfer correlation observed by prior works for standard models actually holds for robust models too. Specifically, we train highly robust ($\varepsilon = 3$)—and thus less accurate—models with six different architectures, and compared ImageNet accuracy against transfer learning performance.

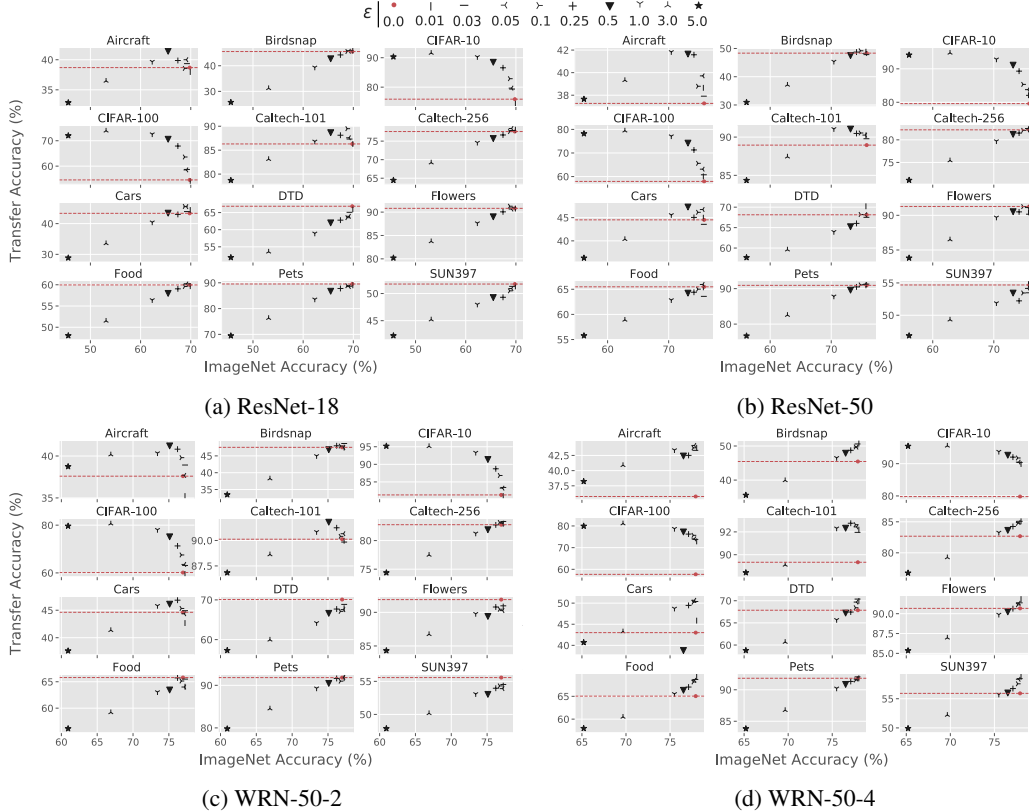

(a) ResNet-18

(b) ResNet-50

(c) WRN-50-2

(d) WRN-50-4

Figure 5: **Fixed-feature** transfer accuracies of standard and robust ImageNet models to various image classification datasets. The linear relationship between ImageNet and transfer accuracies does not hold.

Table 2: Source (ImageNet) and target (CIFAR-10) accuracies, fixing robustness ($\varepsilon$) but varying architecture. When robustness is controlled for, ImageNet accuracy is highly predictive of transfer performance. Similar trends for other datasets are shown in Appendix F.

| Robustness | Dataset | \multicolumn{7}{c}{**Architecture** (see details in Appendix A.1)} |||||||
| | | A | B | C | D | E | F | $R^2$ |
|---|---|---|---|---|---|---|---|---|
| Std ($\varepsilon = 0$) | ImageNet | 77.37 | 77.32 | 73.66 | 65.26 | 64.25 | 60.97 | — |
| | CIFAR-10 | 97.84 | 97.47 | 96.08 | 95.86 | 95.82 | 95.55 | 0.79 |
| Adv ($\varepsilon = 3$) | ImageNet | 66.12 | 65.92 | 56.78 | 50.05 | 42.87 | 41.03 | — |
| | CIFAR-10 | 98.67 | 98.22 | 97.27 | 96.91 | 96.23 | 95.99 | 0.97 |

Table 2 shows that for these models improving ImageNet accuracy improves transfer performance at around the same rate as (and with higher $R^2$ correlation than) standard models.

These observations suggest that transfer learning performance can be further improved by applying known techniques that increase the accuracy of robust models (e.g. [BGH19; Car+19]). More broadly, our findings also indicate that accuracy is not a sufficient measure of feature quality or versatility. Understanding why robust networks transfer particularly well remains an open problem, likely relating to prior work that analyses the features these networks use [Eng+19a; Sha+19; AL20].

### 4.2 Robust models improve with width

Our experiments also reveal a contrast between robust and standard models in how their transfer performance scales with model width. Azizpour et al. [Azi+15], find that although increasing network

depth improves transfer performance, increasing width hurts it. Our results corroborate this trend for standard networks, but indicate that it does *not* hold for robust networks, at least in the regime of widths tested. Indeed, Figure 6 plots results for the three widths of ResNet-50 studied here (x1, x2, and x4), along with a ResNet-18 for reference: as width increases, transfer performance plateaus and decreases for standard models, but continues to steadily grow for robust models. This suggests that scaling network width may further increase the transfer performance gain of robust networks over the standard ones. (This increase comes, however, at a higher computational cost.)

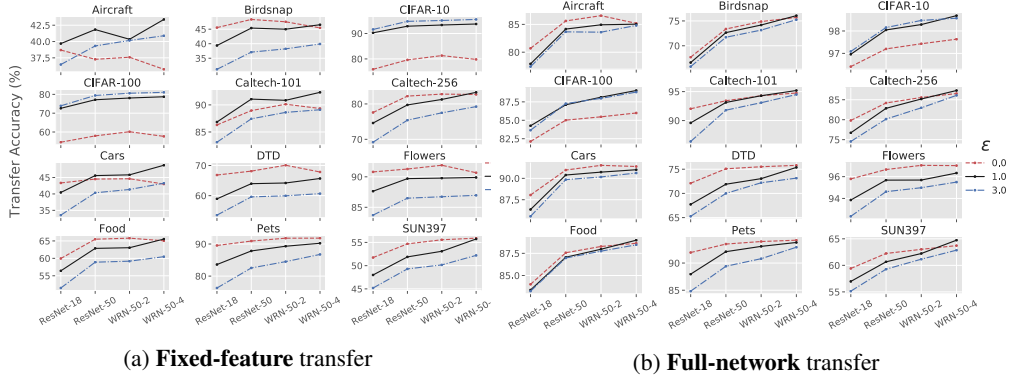

(a) **Fixed-feature** transfer          (b) **Full-network** transfer

Figure 6: Varying width and model robustness while transfer learning from ImageNet to various datasets. Generally, as width increases, transfer learning accuracies of standard models generally plateau or level off while those of robust models steadily increase. More values of $\varepsilon$ are in Appendix F.

## 4.3  Optimal robustness levels for downstream tasks

We observe that although the best robust models often outperform the best standard models, the optimal choice of robustness parameter $\varepsilon$ varies widely between datasets. For example, when transferring to CIFAR-10 and CIFAR-100, the optimal $\varepsilon$ values were 3.0 and 1.0, respectively. In contrast, smaller values of $\varepsilon$ (smaller by an order of magnitude) tend to work better for the rest of the datasets.

One possible explanation for this variability in the optimal choice of $\varepsilon$ might relate to dataset granularity. We hypothesize that on datasets where leveraging finer-grained features are necessary (i.e., where there is less norm-separation between classes in the input space), the most effective values of $\varepsilon$ will be much smaller than for a dataset where leveraging more coarse-grained features suffices. To illustrate this, consider a binary classification task consisting of image-label pairs $(x, y)$, where the correct class for an image $y \in \{0, 1\}$ is determined by a single pixel, i.e., $x_{0,0} = \delta \cdot y$, and $x_{i,j} = 0$, otherwise. We would expect transferring a standard model onto this dataset to yield perfect accuracy regardless of $\delta$, since the dataset is perfectly separable. On the other hand, a robust model is trained to be invariant to perturbations of norm $\varepsilon$—thus, if $\delta < \varepsilon$, the dataset will not appear separable to the standard model and so we expect transfer to be less successful. So, the smaller the $\delta$ (i.e., the larger the "fine grained-ness" of the dataset), the smaller the $\varepsilon$ must be for successful transfer.

**Unifying dataset scale.**  We now present evidence in support of our above hypothesis. Although we lack a quantitative notion of granularity (in reality, features are not simply singular pixels), we consider image resolution as a crude proxy. Since we scale target datasets to match ImageNet dimensions, each pixel in a low-resolution dataset (e.g., CIFAR-10) image translates into several pixels in transfer, thus inflating datasets' separability. Drawing from this observation, we attempt to calibrate the granularities of the 12 image classification datasets used in this work, by first downscaling all the images to the size of CIFAR-10 ($32 \times 32$), and then upscaling them to ImageNet size once more. We then repeat the fixed-feature regression experiments from prior sections, plotting the results in Figure 7 (similar results for full-network transfer are presented in Appendix F). After controlling for original dataset dimension, the datasets' epsilon vs. transfer accuracy curves all behave almost identically to CIFAR-10 and CIFAR-100 ones. Note that while this experimental data supports our hypothesis, we do not take the evidence as an ultimate one and further exploration is needed to reach definitive conclusions.

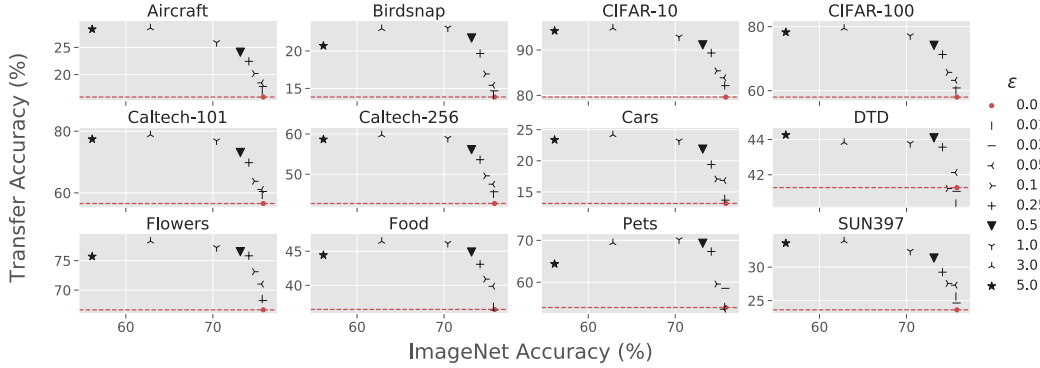

Figure 7: **Fixed-feature** transfer accuracies of various datasets that are down-scaled to $32 \times 32$ before being up-scaled again to ImageNet scale and used for transfer learning. The accuracy curves are closely aligned, unlike those of Figure 5, which illustrates the same experiment without downscaling.

### 4.4 Comparing adversarial robustness to texture robustness

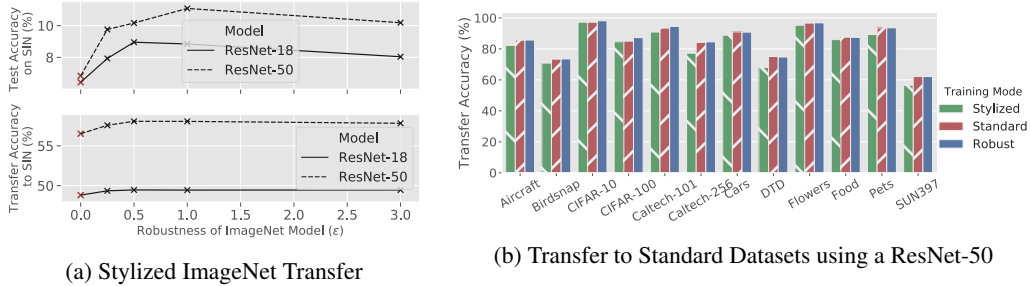

(a) Stylized ImageNet Transfer

(b) Transfer to Standard Datasets using a ResNet-50

Figure 8: We compare standard, stylized and robust ImageNet models on standard transfer tasks (and to stylized ImageNet).

We now investigate the effects of adversarial robustness on transfer learning performance in comparison to other invariances commonly imposed on deep neural networks. Specifically, we consider texture-invariant [Gei+19] models, i.e., models trained on the texture-randomizing Stylized ImageNet (SIN) [Gei+19] dataset. Figure 8b shows that transfer learning from adversarially robust models outperforms transfer learning from texture-invariant models on all considered datasets.

Finally, we use the SIN dataset to further re-inforce the benefits conferred by adversarial robustness. Figure 8a top shows that robust models outperform standard imagenet models when evaluated (top) or fine-tuned (bottom) on Stylized-ImageNet.

## 5 Related Work

A number of works study transfer learning with CNNs [Don+14; Cha+14; Sha+14; Azi+15]. Indeed, transfer learning has been studied in varied domains including medical imaging [MGM18], language modeling [CK18], and various object detection and segmentation related tasks [Ren+15; Dai+16; Hua+17; Che+17]. In terms of methods, others [AGM14; Cha+14; Gir+14; Yos+14; Azi+15; LRM15; HAE16; Chu+16] show that fine-tuning typically outperforms frozen feature-based methods. As discussed throughout this paper, several prior works [Azi+15; HAE16; KSL19; Zam+18; Kol+19; Sun+17; Mah+18; Yos+14] have investigated factors improving or otherwise affecting transfer learning performance. Recently proposed methods have achieved state-of-the-art performance on downstream tasks by scaling up transfer learning techniques [Hua+18; Kol+19].

On the adversarial robustness front, many works—both empirical (e.g., [Mad+18; Miy+18; BGH19; Zha+19]) and certified (e.g., [Lec+19; Wen+18; WK18; RSL18; CRK19; Sal+19; Yan+20])—significantly increase model resilience to adversarial examples [Big+13; Sze+14]. A growing body

of research has studied the *features* learned by these robust networks and suggested that they improve upon those learned by standard networks (cf. [Ily+19; Eng+19a; San+19; AL20; KSJ19; KCL19] and references). On the other hand, prior studies have also identified theoretical and empirical tradeoffs between standard accuracy and adversarial robustness [Tsi+19; BPR18; Su+18; Rag+19]. At the intersection of robustness and transfer learning, Shafahi et al. [Sha+19] investigate transfer learning for increasing downstream-task adversarial robustness (rather than downstream accuracy, as in this work). Aggarwal et al. [Agg+20] find that adversarially trained models perform better at downstream zero-shot learning tasks and weakly-supervised object localization. Finally, concurrent to our work, [Utr+20] also study the transfer performance of adversarially robust networks. Our studies reach similar conclusions and are otherwise complementary: here we study a larger set of downstream datasets and tasks and analyze the effects of model accuracy, model width, and data resolution; Utrera et al. [Utr+20] study the effects of training duration, dataset size, and also introduce an influence function-based analysis [KL17] to study the representations of robust networks. For a detailed discussion of prior work, see Appendix D.

# 6 Conclusion

In this work, we propose using adversarially robust models for transfer learning. We compare transfer learning performance of robust and standard models on a suite of 12 classification tasks, object detection, and instance segmentation. We find that adversarial robust neural networks consistently match or improve upon the performance of their standard counterparts, despite having lower ImageNet accuracy. We also take a closer look at the behavior of adversarially robust networks, and study the interplay between ImageNet accuracy, model width, robustness, and transfer performance.

# Acknowledgements

Work supported in part by the NSF awards CCF-1553428, CNS-1815221, the Open Philanthropy Project AI Fellowship, and the Microsoft Corporation. This material is based upon work supported by the Defense Advanced Research Projects Agency (DARPA) under Contract No. HR001120C0015.

# 7 Statement of Broader Impact

Our work attempts to improve upon standard techniques within computer vision, and as such comes with all of the positive and negative broader impacts of the larger field. More specifically, however, transfer learning allows researchers and practitioners to efficiently train models on their custom datasets starting from models pretrained on large-scale labeled datasets. In this way, transfer learning helps those who are compute-limited or otherwise resource-constrained competititive, and thus makes ML more accessible. We believe that our paper discovers new aspects of pretrained models that make them effective at transfer learning, therefore pushing our understanding of transfer learning and helping us to improve its performance.

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
