[Supplementary Material]

# A Experimental Setup

## A.1 Pretrained ImageNet models

In this paper, we train a number of standard and robust ImageNet models on various architectures. These models are used for all the various transfer learning experiments.

**Architectures** We experiment with several standard architectures from the PyTorch's Torchvision[2]. These models are shown in Tables 3&4.[3]

Table 3: The clean accuracies of standard and $\ell_2$-robust ImageNet classifiers used in our paper.

| | Clean ImageNet Top-1 Accuracy (%) | | | | | | | | | |
|---|---|---|---|---|---|---|---|---|---|---|
| | Robustness parameter $\varepsilon$ | | | | | | | | | |
| **Model** | 0 | 0.01 | 0.03 | 0.05 | 0.1 | 0.25 | 0.5 | 1 | 3 | 5 |
| **ResNet-18** | 69.79 | 69.90 | 69.24 | 69.15 | 68.77 | 67.43 | 65.49 | 62.32 | 53.12 | 45.59 |
| **ResNet-50** | 75.80 | 75.68 | 75.76 | 75.59 | 74.78 | 74.14 | 73.16 | 70.43 | 62.83 | 56.13 |
| **WRN-50-2** | 76.97 | 77.25 | 77.26 | 77.17 | 76.74 | 76.21 | 75.11 | 73.41 | 66.90 | 60.94 |
| **WRN-50-4** | 77.91 | 78.02 | 77.87 | 77.77 | 77.64 | 77.10 | 76.52 | 75.51 | 69.67 | 65.20 |

| | Clean ImageNet Top-1 Accuracy (%) | | | | | |
|---|---|---|---|---|---|---|
| | Model Architecture | | | | | |
| | **A** **DenseNet-161** | **B** **ResNeXt50** | **C** **VGG16-bn** | **D** **MobileNet-v2** | **E** **ShuffleNet** | **F** **MNASNET** |
| $\varepsilon = 0$ | 77.37 | 77.32 | 73.66 | 65.26 | 64.25 | 60.97 |
| $\varepsilon = 3$ | 66.12 | 65.92 | 56.78 | 50.05 | 42.87 | 41.03 |

Table 4: The clean accuracies of $\ell_\infty$-robust ImageNet classifiers.

| | Clean ImageNet Top-1 Accuracy (%) | | | | |
|---|---|---|---|---|---|
| | Robustness parameter $\varepsilon$ | | | | |
| **Model** | $\frac{0.5}{255}$ | $\frac{1}{255}$ | $\frac{2}{255}$ | $\frac{4}{255}$ | $\frac{8}{255}$ |
| **ResNet-18** | 66.13 | 63.46 | 59.63 | 52.49 | 42.11 |
| **ResNet-50** | 73.73 | 72.05 | 69.10 | 63.86 | 54.53 |
| **WRN-50-2** | 75.82 | 74.65 | 72.35 | 68.41 | 60.82 |

**Training details** We fix the training procedure for all of these models. We train all the models from scratch using SGD with batch size of 512, momentum of 0.9, and weight decay of $1e-4$. We train for 90 epochs with an initial learning rate of 0.1 that drops by a factor of 10 every 30 epochs.

For **Standard Training**, we use the standard cross-entropy multi-class classification loss. For **Robust Training**, we use adversarial training [Mad+18]. We train on adversarial examples generated within maximum allowed perturbations $\ell_2$ of $\varepsilon \in \{0.01, 0.03, 0.05, 0.1, 0.25, 0.5, 1, 3, 5\}$ and $\ell_\infty$ perturbations of $\varepsilon \in \{\frac{0.5}{255}, \frac{1}{255}, \frac{2}{255}, \frac{4}{255}, \frac{8}{255}\}$ using 3 attack steps and a step size of $\frac{\varepsilon \times 2}{3}$.

Table 5: Classification datasets used in this paper.

| Dataset | Classes | Size (Train/Test) | Accuracy Metric |
|---|---|---|---|
| Birdsnap [Ber+14] | 500 | 32,677/8,171 | Top-1 |
| Caltech-101 [FFP04] | 101 | 3,030/5,647 | Mean Per-Class |
| Caltech-256 [GHP07] | 257 | 15,420/15,187 | Mean Per-Class |
| CIFAR-10 [Kri09] | 10 | 50,000/10,000 | Top-1 |
| CIFAR-100 [Kri09] | 100 | 50,000/10,000 | Top-1 |
| Describable Textures (DTD) [Cim+14] | 47 | 3,760/1,880 | Top-1 |
| FGVC Aircraft [Maj+13] | 100 | 6,667/3,333 | Mean Per-Class |
| Food-101 [BGV14] | 101 | 75,750/25,250 | Top-1 |
| Oxford 102 Flowers [NZ08] | 102 | 2,040/6,149 | Mean Per-Class |
| Oxford-IIIT Pets [Par+12] | 37 | 3,680/3,669 | Mean Per-Class |
| SUN397 [Xia+10] | 397 | 19,850/19,850 | Top-1 |
| Stanford Cars [Kra+13] | 196 | 8,144/8,041 | Top-1 |

## A.2 ImageNet transfer to classification datasets

### A.2.1 Datasets

We test transfer learning starting from ImageNet pretrained models on classification datasets that are used in [KSL19]. These datasets vary in size the number of classes and datapoints. The details are shown in Table 5.

### A.2.2 Fixed-feature Transfer

For this type of transfer learning, we *freeze* the weights of the ImageNet pretrained model[4], and replace the last fully connected layer with a random initialized one that fits the transfer dataset. We train only this new layer for 150 epochs using SGD with batch size of 64, momentum of 0.9, weight decay of $5e - 4$, and an initial lr $\in \{0.01, 0.001\}$ that drops by a factor of 10 every 50 epochs. We use the following standard data-augmentation methods:

```
TRAIN_TRANSFORMS = transforms.Compose([
    transforms.RandomResizedCrop(224),
    transforms.RandomHorizontalFlip(),
    transforms.ToTensor(),
])
TEST_TRANSFORMS = transforms.Compose([
    transforms.Resize(256),
    transforms.CenterCrop(224),
    transforms.ToTensor()
])
```

### A.2.3 Full-network transfer

For full-network transfer learning, we use the exact same hyperparameters as the fixed-feature setting, but we *do not freeze* the weights of the pretrained ImageNet model.

## A.3 Unifying dataset scale

For this experiment, we follow the exact experimental setup of A.2 with the only modification being resizing all the datasets to $32 \times 32$ then do dataaugmentation as before:

```
TRAIN_TRANSFORMS = transforms.Compose([
    transforms.Resize(32),
    transforms.RandomResizedCrop(224),
```

```
        transforms.RandomHorizontalFlip(),
        transforms.ToTensor(),
    ])
    TEST_TRANSFORMS = transforms.Compose([
        transforms.Resize(32),
        transforms.Resize(256),
        transforms.CenterCrop(224),
        transforms.ToTensor()
    ])
```

### A.4  Replicate our results

We desired simplicity and kept reproducibility in our minds when conducting our experiments, so we use standard hyperparameters and minimize the number of tricks needed to replicate our results. We open source all the standard and robust ImageNet models that we use in our paper, and our code is available at https://github.com/Microsoft/robust-models-transfer.

## B  Transfer Learning with $\ell_\infty$-robust ImageNet models

We investigate how well other types of robust ImageNet models do in transfer learning.

Table 6: Transfer Accuracy of standard vs $\ell_\infty$-robust ImageNet models on CIFAR-10 and CIFAR-100.

| | | | Transfer Accuracy (%) | | | | | |
|---|---|---|---|---|---|---|---|---|
| | | | Robustness parameter $\varepsilon$ | | | | | |
| Dataset | Transfer Type | Model | 0.0 | $\frac{0.5}{255}$ | $\frac{1.0}{255}$ | $\frac{2.0}{255}$ | $\frac{4.0}{255}$ | $\frac{8.0}{255}$ |
| CIFAR-10 | Full-network | ResNet-18 | 96.05 | 96.85 | 96.80 | 96.98 | **97.04** | 96.79 |
| | | ResNet-50 | 97.14 | 97.69 | 97.84 | 97.98 | 97.92 | 98.01 |
| | Fixed-feature | ResNet-18 | 75.02 | 87.13 | 89.01 | 89.07 | **90.56** | 89.18 |
| | | ResNet-50 | 78.16 | 90.55 | 91.51 | 92.74 | 93.35 | 93.68 |
| CIFAR-100 | Full-network | ResNet-18 | 81.70 | 83.66 | 83.46 | **83.98** | 83.55 | 82.82 |
| | | ResNet-50 | 84.75 | 86.12 | 86.48 | 87.06 | 86.90 | 86.21 |
| | Fixed-feature | ResNet-18 | 53.86 | 68.52 | 70.83 | 72.00 | **72.19** | 69.78 |
| | | ResNet-50 | 55.57 | 72.89 | 74.16 | 76.22 | 77.17 | 76.70 |

## C  Object Detection and Instance Segmentation

In this section we provide more experimental details, and results, relating to our object detection and instance segmentation experiments.

**Experimental setup.**  We use only standard configurations from Detectron2[5] to train models. For COCO tasks, compute limitations made training from every $\varepsilon$ initialization impossible. Instead, we trained from every $\varepsilon$ initialization using a reduced learning rate schedule (the corresponding 1x learning rate schedule in Detectron2) before training from the top three $\varepsilon$ initializations (by Box AP) along with the standard model using the full learning rate training schedule (the 3x schedule). Our results for the 1x learning rate search are in Figure 9; our results, similar to those in Section 3.2, show that training from a robustly trained backbone yields greater AP than training from a standard-trained backbone.

Figure 9: AP of instance segmentation and object detection models with backbones initialized with $\varepsilon$-robust models before training. Robust backbones generally lead to better AP, and the best robust backbone always outperforms the standard-trained backbone for every task.

**Baselines.** We use standard ResNet-50 models from the torchvision package[6] using the Robustness library [Eng+19b]. Detectron2 models were originally trained for (and their configurations are tuned for) ResNet-50 models from the original ResNet code release[7], which are slightly different from the torchvision ResNet-50s we use. It has been previously noted that models trained from torchvision perform worse with Detectron2 than these original models[8]. Despite this, the best torchvision ResNet-50 models we train from robust initializations dominate (without any additional hyperparameter searching) the original baselines except for the COCO Object Detection task in terms of AP, in which the original baseline has 0.07 larger Box AP[9].

# D Related Work

In this section, we describe some of the related work to our paper.

## D.1 Transfer learning

Transfer learning has been investigated in a number of early works which extracted features from ImageNet CNNs and trained SVMs or logistic regression classifiers using these features on new datasets/tasks [Don+14; Cha+14; Sha+14]. These ImageNet features were shown to outperform hand-crafted features even on tasks different from ImageNet classification.[Sha+14; Don+14]. Later on, [Azi+15] demonstrated that transfer using deep networks is more effective than using wide networks across many transfer tasks. A number of works has furthermore studied the transfer problem in the domain of medical imaging [MGM18] and language modeling [CK18]. Besides, many of research in the literature has indicated that, specifically in computer vision, fine-tuning typically performs better than than classification based on freezed features [AGM14; Cha+14; Gir+14; Yos+14; Azi+15; LRM15; HAE16; Chu+16].

ImageNet pretrained networks have also been widely used as backbone models for various object detections models including Faster R-CNN and R-FCN [Ren+15; Dai+16]. More accurate ImageNet models tend to lead to better overall object detection accuracy [Hua+17]. Similar usage is also common in image segmentation [Che+17].

Several works have studied how modifying the source dataset can affect the transfer accuracy. [Azi+15; HAE16] investigated the importance of the number of classes vs. number of images per class for learning better fixed image features, and these works have reached to conflicting conclusions [KSL19]. [Yos+14] showed that freezing only the first layer of AlexNet does not affect the transfer performance between natural and manmade subsets of ImageNet as opposed to freezing more layers. Other works demonstrated that transfer learning works even when the target dataset is large by transferring features learnt on a very large image datasets to ImageNet [Sun+17; Mah+18].

More recently, [Zam+18] proposed a method to improve the efficiency of transfer learning when labeled data from multiple domain are available. Furthermore, studied whether better ImageNet models transfer better to other datasets or not [KSL19]. It shows a strong correlation between the transfer accuracy of a pretrained ImageNet model (both for the logistic regression and finetuning settings) and the top-1 accuracy of these models on ImageNet. Finally, [Kol+19] explored pre-training using enormous amount of data of around 300 million noisily labelled images, and showed improvements in transfer learning over pretraining on ImageNet for several tasks.

## D.2 Transfer learning and robustness

A recent work [Sha+19] investigated the problem of adversarially robust transfer learning: transferring adversarially robust representations to new datasets while maintaining robustness on the downstream task. While this work might look very similar to ours, there are two key differences. The first is that this work investigates using robust source models for the purpose of improving/maintaining robustness on the downstream task, did not investigate whether robust source models can improve the clean accuracy on the downstream tasks. The second is that they point out that starting from a standard trained ImageNet model leads to better natural accuracy when used for downstream tasks, the opposite of what we show in the paper: we show that one can get better transfer accuracies using robust, but less accurate, ImageNet pretrained models.

## D.3 Robustness as a prior for learning representation

A major goal in deep learning is to learn robust high-level feature representations of input data. However, current standard neural networks seem to learn non-robust features that can be easily exploited to generate adversarial examples. On the other hand, a number of recent papers have argued that the features learned by adversarially robust models are less vulnerable to adversarial examples, and at the same time are more perceptually aligned with humans [Ily+19; Eng+19a]. Specifically, [Ily+19] presented a framework to study and disentangle robust and non-robust features for standard trained networks. Concurrently, [Eng+19a] utilized this framework to show that robust optimization can be re-cast as a tool for enforcing priors on the features learned by deep neural

networks. They showed that the representations learned by robust models make significant progress towards learning a high-level encoding of inputs.

## E  Background on Adversarially Robust Models

**Adversarial examples in computer vision.**  Adversarial examples [Big+13; Sze+14] (also referred to as *adversarial attacks*) are imperceptible perturbations to natural inputs that induce misbehaviour from machine learning—in this context computer vision—systems. An illustration of such an attack is shown in Figure 10. The discovery of adversarial examples was a major contributor to the rise of *deep learning security*, where prior work has focused on both robustifying models against such attacks (cf. [GSS15; Mad+18; WK18; RSL18; CRK19] and their references), as well as testing the robustness of machine learning systems in "real-world" settings (cf. [Pap+17; Ath+18; Ily+18; LSK19; Evt+18] and their references). A model that is resilient to such adversarial examples is referred to as "adversarially robust."

**Robust optimization and adversarial training.**  One of the canonical methods for training an adversarially robust model is robust optimization. Typically, we train deep learning models using empirical risk minimization (ERM) over the training data—that is, we solve:

$$\min_{\theta} \frac{1}{n} \sum_{i=1}^{n} \mathcal{L}(x_i, y_i; \theta),$$

where $\theta$ represents the model parameters, $\mathcal{L}$ is a task-dependent loss function (e.g., cross-entropy loss for classification), and $\{(x_i, y_i) \sim \mathcal{D}\}$ are training image-label pairs. In robust optimization (dating back to the work of Wald [Wal45]), we replace this standard ERM objective with a *robust* risk minimization objective:

$$\min_{\theta} \frac{1}{n} \sum_{i=1}^{n} \max_{x'; d(x_i, x') < \varepsilon} \mathcal{L}(x', y_i),$$

where $d$ is a fixed but arbitrary norm. (In practice, $d$ is often assumed to be an $\ell_p$ norm for $p \in \{2, \infty\}$—for the majority of this work we set $p = 2$, so $d(x, x')$ is the Euclidean norm.) In short, rather than minimizing the loss on only the training points, we instead minimize the worst-case loss over a ball around each training point. Assuming the robust objective generalizes, it ensures that an adversary cannot perturb a given test point $(x, y) \sim \mathcal{D}$ and drastically increase the loss of the model. The parameter $\varepsilon$ governs the desired robustness of the model: $\varepsilon = 0$ corresponds to standard (ERM) training, and increasing $\varepsilon$ results in models that are stable within larger and larger radii.

Figure 10: An example of an adversarial attack: adding the imperceptible perturbation (middle) to a correctly classified pig (left) results in a near-identical image that is classified as "airliner" by an Inception-v3 ImageNet model.

At first glance, it is unclear how to effectively solve the robust risk minimization problem posed above—typically we use SGD to minimize risk, but here the loss function has an embedded maximization, so the corresponding SGD update rule would be:

$$\theta_t \leftarrow \theta_{t-1} - \eta \cdot \nabla_\theta \left( \max_{x'; d(x', x_i) < \varepsilon} \mathcal{L}(x', y_i; \theta) \right).$$

Thus, to actually train an adversarially robust neural network, Madry et al. [Mad+18] turn to inspiration from robust convex optimization, where Danskin's theorem [Dan67] says that for a function $f(\alpha, \beta)$ that is convex in $\alpha$,

$$\nabla_\alpha \left( \max_{\beta \in B} f(\alpha, \beta) \right) = \nabla_\alpha f(\alpha, \beta^*), \qquad \text{where } \beta^* = \arg\max_\beta f(\alpha, \beta) \text{ and } B \text{ is compact.}$$

Danskin's theorem thus allows us to write the gradient of a minimax problem in terms of only the gradient of the inner objective, evaluated at its maximal point. Carrying this intuition over to the neural network setting (despite the lack of convexity) results in the popular *adversarial training* algorithm [GSS15; Mad+18], where at each training iteration, worst-case (adversarial) inputs are passed to the neural network rather than standard unmodified inputs. Despite its simplicity, adversarial training remains a competitive baseline for training adversarially robust networks [RWK20]. Furthermore, recent works have provided theoretical evidence for the success of adversarial training directly in the neural network setting [Gao+19; AL20; Zha+20].

# F   Omitted Figures

## F.1   More Runs for the Main Classification Results: additional results to 2 & 3

Figure 11: **Fixed-feature** transfer learning results using standard and robust models for the 12 downstream image classification tasks considered. Error bars denote the standard deviation over **ten random trials**.

Figure 12: **Full-network** transfer learning results using standard and robust models for the 12 downstream image classification tasks considered. Error bars denote the standard deviation over **ten random trials**.

## F.2  Full-network Transfer: additional results to Figure 5

(a) ResNet-18

(b) ResNet-50

(c) WRN-50-2

(d) WRN-50-4

Figure 13: **Full-network** transfer accuracies of standard and robust ImageNet models to various image classification datasets. The linear relationship between accuracy and transfer performance does not hold; instead, for fixed accuracy, generally increased robustness yields higher transfer accuracy.

## F.3 Varying architecture: additional results to Table 2

Table 7: Source (ImageNet) and target accuracies, fixing robustness ($\varepsilon$) but varying architecture. When robustness is controlled for, ImageNet accuracy is highly predictive of (full-network) transfer performance.

| Robustness | Dataset | Architecture (see details in Appendix A.1) | | | | | | $R^2$ |
|---|---|---|---|---|---|---|---|---|
| | | A | B | C | D | E | F | |
| Std ($\varepsilon = 0$) | ImageNet | 77.37 | 77.32 | 73.66 | 65.26 | 64.25 | 60.97 | — |
| | CIFAR-10 | 97.84 | 97.47 | 96.08 | 95.86 | 95.82 | 95.55 | 0.79 |
| | CIFAR-100 | 86.53 | 85.53 | 82.07 | 80.02 | 80.76 | 80.41 | 0.82 |
| | Caltech-101 | 94.78 | 94.63 | 91.32 | 88.91 | 87.13 | 83.28 | 0.94 |
| | Caltech-256 | 86.22 | 86.33 | 82.23 | 76.51 | 75.81 | 74.90 | 0.98 |
| | Cars | 91.28 | 91.27 | 90.97 | 88.31 | 85.81 | 84.54 | 0.91 |
| | Flowers | 97.93 | 97.29 | 96.80 | 96.25 | 95.40 | 72.06 | 0.44 |
| | Pets | 94.55 | 94.26 | 92.63 | 89.78 | 88.59 | 82.69 | 0.87 |
| Adv ($\varepsilon = 3$) | ImageNet | 66.12 | 65.92 | 56.78 | 50.05 | 42.87 | 41.03 | — |
| | CIFAR-10 | 98.67 | 98.22 | 97.27 | 96.91 | 96.23 | 95.99 | 0.97 |
| | CIFAR-100 | 88.65 | 88.32 | 84.14 | 83.32 | 80.92 | 80.52 | 0.97 |
| | Caltech-101 | 93.84 | 93.31 | 89.93 | 89.02 | 83.29 | 75.52 | 0.83 |
| | Caltech-256 | 84.35 | 83.05 | 78.19 | 74.08 | 69.19 | 70.04 | 0.99 |
| | Cars | 90.91 | 90.08 | 89.67 | 88.02 | 83.57 | 78.76 | 0.79 |
| | Flowers | 95.77 | 96.01 | 93.88 | 94.25 | 91.47 | 26.98 | 0.38 |
| | Pets | 91.85 | 91.46 | 88.06 | 85.63 | 80.92 | 64.90 | 0.72 |

## F.4 Stylized ImageNet Transfer: additional results to Figure 8b

(a) **Fixed-feature** ResNet-18

(b) **Fixed-feature** ResNet-50

(c) **Full-network** ResNet-18

(d) **Full-network** ResNet-50

Figure 14: We compare standard, stylized and robust ImageNet models on standard transfer tasks.

## F.5 Unified scale: additional results to Figure 7

(a) ResNet-18

(b) ResNet-50 (same as Figure 7)

Figure 15: **Fixed-feature** transfer accuracies of various datasets that are down-scaled to $32 \times 32$ before being up-scaled again to ImageNet scale and used for transfer learning. The accuracy curves are closely aligned, unlike those of Figure 5, which illustrates the same experiment without down-scaling.

(a) ResNet-18

(b) ResNet-50

Figure 16: **Full-network** transfer accuracies of various datasets that are down-scaled to $32 \times 32$ before being up-scaled again to ImageNet scale and used for transfer learning.

## F.6 Effect of width: additional results to Figure 6

(a) **Fixed-feature** transfer

(b) **Full-network** transfer

Figure 17: Varying width and model robustness while transfer learning from ImageNet to various datasets. Generally, as width increases, transfer learning accuracies of standard models generally plateau or level off while those of robust models steadily increase.

# G   Detailed Numerical Results

## G.1   Fixed-feature transfer to classification tasks (Fig. 5)

Table 8: **Fixed-feature** transfer for various standard and robust ImageNet models and datasets.

| | | Transfer Accuracy (%) | | | | | | | | | |
|---|---|---|---|---|---|---|---|---|---|---|---|
| | | Robustness parameter $\varepsilon$ | | | | | | | | | |
| Dataset | Model | 0.00 | 0.01 | 0.03 | 0.05 | 0.10 | 0.25 | 0.50 | 1.00 | 3.00 | 5.00 |
| Aircraft | ResNet-18 | 38.69 | 37.96 | 39.35 | 40.00 | 38.55 | 39.87 | **41.40** | 39.68 | 36.47 | 32.87 |
| | ResNet-50 | 37.27 | 38.65 | 37.91 | 39.71 | 38.79 | 41.58 | 41.64 | **41.83** | 39.32 | 37.65 |
| | WRN-50-2 | 37.59 | 35.22 | 38.92 | 37.68 | 39.80 | 40.81 | **41.20** | 40.34 | 40.16 | 38.74 |
| | WRN-50-4 | 35.74 | 43.76 | 43.34 | **44.14** | 43.75 | 42.51 | 42.40 | 43.38 | 40.88 | 38.23 |
| Birdsnap | ResNet-18 | 45.54 | **45.88** | 45.86 | 45.66 | 45.55 | 44.23 | 42.72 | 39.38 | 31.19 | 25.73 |
| | ResNet-50 | 48.35 | 48.86 | 47.84 | 48.24 | **49.19** | 48.73 | 47.48 | 45.38 | 37.10 | 30.95 |
| | WRN-50-2 | 47.54 | 47.47 | **48.68** | 47.48 | 47.93 | 48.01 | 46.84 | 44.99 | 38.23 | 33.47 |
| | WRN-50-4 | 45.45 | **50.72** | 50.60 | 49.66 | 49.73 | 48.73 | 47.88 | 46.53 | 39.91 | 35.58 |
| CIFAR-10 | ResNet-18 | 75.91 | 74.33 | 79.35 | 79.67 | 82.87 | 86.58 | 88.45 | 90.27 | **91.59** | 90.31 |
| | ResNet-50 | 79.61 | 82.12 | 82.07 | 83.78 | 85.35 | 89.31 | 91.10 | 92.86 | **94.77** | 94.16 |
| | WRN-50-2 | 81.31 | 80.98 | 83.43 | 83.23 | 86.83 | 88.73 | 91.37 | 93.34 | 95.12 | **95.19** |
| | WRN-50-4 | 79.81 | 89.90 | 90.35 | 90.48 | 91.76 | 92.03 | 92.62 | 93.73 | **95.53** | 95.43 |
| CIFAR-100 | ResNet-18 | 54.58 | 53.92 | 58.70 | 58.51 | 63.60 | 67.91 | 70.58 | 72.60 | **73.91** | 72.01 |
| | ResNet-50 | 57.94 | 60.06 | 60.76 | 63.13 | 65.61 | 71.29 | 74.18 | 77.14 | **79.43** | 78.20 |
| | WRN-50-2 | 60.14 | 59.52 | 63.12 | 63.55 | 67.51 | 71.30 | 75.11 | 78.07 | **80.61** | 79.64 |
| | WRN-50-4 | 57.68 | 72.88 | 73.79 | 74.06 | 75.68 | 76.25 | 77.23 | 78.73 | **81.08** | 79.94 |
| Caltech-101 | ResNet-18 | 86.30 | 86.28 | 87.32 | 87.59 | **89.49** | 88.12 | 88.65 | 86.84 | 83.11 | 78.69 |
| | ResNet-50 | 88.95 | 90.22 | 89.79 | 90.26 | 90.54 | 90.48 | 91.04 | **91.07** | 87.43 | 84.35 |
| | WRN-50-2 | 90.12 | 89.97 | 89.85 | 90.67 | 90.40 | 91.25 | **91.80** | 90.84 | 88.62 | 86.83 |
| | WRN-50-4 | 89.34 | 92.20 | 91.96 | 92.44 | 92.63 | **92.76** | 92.32 | 92.32 | 89.10 | 88.43 |
| Caltech-256 | ResNet-18 | 77.58 | 78.09 | 77.87 | **78.40** | 77.57 | 76.66 | 75.69 | 74.61 | 69.19 | 64.46 |
| | ResNet-50 | 82.21 | 82.31 | 82.23 | **82.51** | 82.10 | 81.50 | 81.21 | 79.72 | 75.42 | 71.07 |
| | WRN-50-2 | 82.78 | 82.94 | **83.34** | 83.04 | 83.17 | 82.74 | 81.89 | 81.26 | 77.48 | 74.38 |
| | WRN-50-4 | 82.68 | 85.07 | **85.08** | 84.88 | 84.75 | 84.24 | 83.62 | 83.27 | 79.24 | 76.75 |
| Cars | ResNet-18 | 43.34 | 44.43 | 43.92 | 45.53 | **45.59** | 43.00 | 43.40 | 40.45 | 33.55 | 28.86 |
| | ResNet-50 | 44.52 | 44.98 | 43.56 | 46.74 | 46.15 | 45.04 | **47.28** | 45.58 | 40.34 | 36.32 |
| | WRN-50-2 | 44.63 | 42.67 | 44.92 | 44.36 | 45.32 | **46.83** | 46.10 | 45.81 | 41.35 | 37.62 |
| | WRN-50-4 | 43.01 | 45.86 | 50.39 | **50.67** | 50.22 | 49.46 | 38.77 | 48.73 | 43.26 | 40.68 |
| DTD | ResNet-18 | **66.84** | 66.01 | 65.07 | 63.90 | 63.51 | 62.78 | 61.99 | 58.94 | 53.55 | 51.88 |
| | ResNet-50 | 68.14 | **70.21** | 67.52 | 68.16 | 68.21 | 66.03 | 65.21 | 63.97 | 59.59 | 57.68 |
| | WRN-50-2 | **70.09** | 67.89 | 68.87 | 67.55 | 67.11 | 67.70 | 66.61 | 64.20 | 59.95 | 57.29 |
| | WRN-50-4 | 67.85 | 69.95 | **70.37** | 69.70 | 68.42 | 67.45 | 67.22 | 65.69 | 60.67 | 58.78 |
| Flowers | ResNet-18 | 90.80 | 90.76 | 90.88 | 90.65 | **91.26** | 90.05 | 88.99 | 87.64 | 83.72 | 80.20 |
| | ResNet-50 | **91.28** | 90.43 | 90.16 | 91.12 | 91.26 | 90.50 | 90.52 | 89.70 | 86.49 | 83.85 |
| | WRN-50-2 | **91.90** | 90.86 | 90.97 | 90.26 | 90.46 | 90.79 | 89.39 | 89.79 | 86.73 | 84.31 |
| | WRN-50-4 | 90.67 | **91.84** | 91.37 | 91.32 | 91.12 | 90.63 | 90.23 | 89.89 | 86.96 | 85.35 |
| Food | ResNet-18 | 59.96 | 59.67 | **60.20** | 60.17 | 59.59 | 59.04 | 57.97 | 56.42 | 51.49 | 48.03 |
| | ResNet-50 | 65.49 | 65.39 | 63.59 | **65.95** | 65.02 | 64.41 | 64.23 | 62.86 | 58.90 | 55.77 |
| | WRN-50-2 | **65.80** | 64.06 | 65.50 | 64.00 | 65.14 | 65.73 | 63.44 | 63.05 | 59.19 | 56.13 |
| | WRN-50-4 | 65.04 | **69.26** | 68.69 | 68.50 | 68.15 | 67.03 | 66.32 | 65.53 | 60.48 | 57.98 |
| Pets | ResNet-18 | **89.55** | 89.03 | 88.67 | 88.54 | 88.87 | 87.80 | 86.73 | 83.61 | 76.29 | 69.48 |
| | ResNet-50 | 90.92 | 90.93 | **91.27** | 91.16 | 91.05 | 90.48 | 89.57 | 87.84 | 82.54 | 76.69 |
| | WRN-50-2 | 91.81 | 91.69 | 91.83 | **91.85** | 90.98 | 91.61 | 90.46 | 89.31 | 84.51 | 79.80 |
| | WRN-50-4 | 91.83 | 91.82 | **92.05** | 91.70 | 91.54 | 91.32 | 90.85 | 90.23 | 86.75 | 83.83 |
| SUN397 | ResNet-18 | **51.74** | 51.31 | 51.32 | 50.92 | 50.50 | 49.30 | 49.25 | 47.99 | 45.19 | 42.24 |
| | ResNet-50 | 54.69 | **54.82** | 53.48 | 54.15 | 53.45 | 52.23 | 53.43 | 51.88 | 49.30 | 46.84 |
| | WRN-50-2 | **55.57** | 54.35 | 54.53 | 53.90 | 54.31 | 53.96 | 53.03 | 53.09 | 50.16 | 47.86 |
| | WRN-50-4 | 55.92 | **58.75** | 58.45 | 58.34 | 57.56 | 56.75 | 55.99 | 55.74 | 52.21 | 49.91 |

## G.2 Full-network transfer to classification tasks (Fig. 3)

Table 9: **Full-network** transfer for various standard and robust ImageNet models and datasets.

| | | Transfer Accuracy (%) | | | | | | | | | |
|---|---|---|---|---|---|---|---|---|---|---|---|
| | | Robustness parameter $\varepsilon$ | | | | | | | | | |
| Dataset | Model | 0.00 | 0.01 | 0.03 | 0.05 | 0.10 | 0.25 | 0.50 | 1.00 | 3.00 | 5.00 |
| Aircraft | ResNet-18 | **80.70** | 80.32 | 79.99 | 80.06 | 79.30 | 78.74 | 77.69 | 77.90 | 77.41 | 77.26 |
| | ResNet-50 | 85.62 | 85.62 | 85.61 | **85.72** | 84.73 | 84.65 | 84.77 | 84.16 | 83.66 | 83.77 |
| | WRN-50-2 | **86.57** | 86.08 | 85.81 | 86.06 | 85.17 | 85.60 | 85.55 | 84.93 | 83.60 | 83.80 |
| | WRN-50-4 | 85.19 | 85.98 | 86.10 | 86.11 | **86.24** | 85.88 | 85.67 | 85.04 | 84.81 | 85.43 |
| Birdsnap | ResNet-18 | 67.71 | **67.96** | 67.58 | 67.86 | 67.80 | 67.63 | 67.10 | 66.62 | 65.80 | 64.81 |
| | ResNet-50 | 73.38 | **73.52** | 73.39 | 73.33 | 73.22 | 73.48 | 73.21 | 72.65 | 71.71 | 71.05 |
| | WRN-50-2 | 74.87 | **74.98** | 74.85 | 74.93 | 74.75 | 74.80 | 74.79 | 74.18 | 73.15 | 72.64 |
| | WRN-50-4 | 75.71 | **76.55** | 76.47 | 76.14 | 76.18 | 76.29 | 76.20 | 76.06 | 75.25 | 74.40 |
| CIFAR-10 | ResNet-18 | 96.41 | 96.30 | 96.46 | 96.47 | 96.67 | 96.83 | 97.04 | 96.96 | **97.09** | 96.92 |
| | ResNet-50 | 97.20 | 97.26 | 97.52 | 97.43 | 97.59 | 97.71 | 97.86 | 98.05 | **98.15** | **98.15** |
| | WRN-50-2 | 97.43 | 97.60 | 97.72 | 97.69 | 97.86 | 98.02 | 98.09 | 98.29 | **98.47** | 98.44 |
| | WRN-50-4 | 97.63 | 98.51 | 98.52 | 98.59 | 98.62 | 98.52 | 98.55 | **98.68** | 98.57 | 98.53 |
| CIFAR-100 | ResNet-18 | 82.13 | 82.36 | 82.82 | 82.71 | 83.14 | 83.85 | 84.19 | **84.25** | 83.65 | 83.36 |
| | ResNet-50 | 85.02 | 85.20 | 85.45 | 85.44 | 85.80 | 86.31 | 86.64 | 87.10 | **87.26** | 86.43 |
| | WRN-50-2 | 85.47 | 85.94 | 85.95 | 86.15 | 86.47 | 87.31 | 87.52 | **88.13** | 87.98 | 87.54 |
| | WRN-50-4 | 85.99 | 88.70 | 88.61 | 88.72 | 88.72 | 88.75 | 88.80 | **89.04** | 88.83 | 88.62 |
| Caltech-101 | ResNet-18 | **92.04** | 90.81 | 91.28 | 91.29 | 89.75 | 90.73 | 91.12 | 89.60 | 86.39 | 86.95 |
| | ResNet-50 | 93.42 | 93.82 | **94.53** | 94.18 | 94.27 | 94.24 | 93.79 | 93.13 | 91.79 | 89.97 |
| | WRN-50-2 | 94.29 | 94.43 | 94.13 | 94.49 | 94.48 | 94.92 | **95.29** | 94.28 | 93.08 | 91.89 |
| | WRN-50-4 | 94.76 | 95.60 | 95.32 | **95.62** | 95.30 | 95.45 | 95.23 | 95.19 | 94.49 | 93.25 |
| Caltech-256 | ResNet-18 | 79.80 | 80.00 | 79.45 | **80.10** | 79.23 | 79.07 | 78.86 | 76.71 | 74.55 | 71.57 |
| | ResNet-50 | 84.19 | 84.30 | 84.37 | **84.54** | 84.04 | 84.12 | 84.02 | 82.85 | 80.15 | 77.81 |
| | WRN-50-2 | 85.56 | 85.65 | 86.04 | **86.26** | 85.91 | 85.67 | 85.80 | 85.19 | 82.97 | 81.04 |
| | WRN-50-4 | 86.56 | 87.53 | 87.54 | **87.62** | **87.62** | 87.54 | 87.38 | 87.31 | 86.09 | 84.08 |
| Cars | ResNet-18 | **88.05** | 87.80 | 87.53 | 87.90 | 87.45 | 87.10 | 86.94 | 86.35 | 85.56 | 85.26 |
| | ResNet-50 | **90.97** | 90.65 | 90.83 | 90.52 | 90.23 | 90.47 | 90.59 | 90.39 | 89.85 | 89.28 |
| | WRN-50-2 | **91.52** | 91.47 | 91.27 | 91.20 | 91.04 | 91.06 | 91.05 | 90.73 | 90.16 | 90.27 |
| | WRN-50-4 | **91.39** | 91.09 | 91.14 | 91.05 | 91.10 | 91.03 | 91.12 | 91.01 | 90.63 | 90.34 |
| DTD | ResNet-18 | **72.11** | 71.37 | 71.54 | 70.73 | 70.37 | 70.07 | 68.46 | 67.73 | 65.27 | 65.41 |
| | ResNet-50 | **75.09** | 74.77 | 74.54 | 74.02 | 73.56 | 72.89 | 73.19 | 71.90 | 70.00 | 70.02 |
| | WRN-50-2 | 75.51 | **75.94** | 75.41 | 74.98 | 74.65 | 74.57 | 74.95 | 73.05 | 72.20 | 71.31 |
| | WRN-50-4 | 75.80 | 76.65 | **76.93** | 76.47 | 76.44 | 76.54 | 75.57 | 75.37 | 73.16 | 72.84 |
| Flowers | ResNet-18 | **95.79** | 95.31 | 95.20 | 95.44 | 95.49 | 94.82 | 94.53 | 93.86 | 92.36 | 91.42 |
| | ResNet-50 | 96.65 | **96.81** | 96.50 | 96.53 | 96.20 | 96.25 | 95.99 | 95.68 | 94.62 | 94.20 |
| | WRN-50-2 | 97.04 | **97.21** | 96.71 | 96.74 | 96.63 | 96.35 | 96.07 | 95.69 | 94.98 | 94.67 |
| | WRN-50-4 | **97.01** | 96.52 | 96.59 | 96.53 | 96.53 | 96.38 | 96.28 | 96.33 | 95.50 | 94.92 |
| Food | ResNet-18 | **84.01** | 83.95 | 83.74 | 83.69 | 83.89 | 83.78 | 83.60 | 83.36 | 83.23 | 82.91 |
| | ResNet-50 | **87.57** | 87.42 | 87.45 | 87.46 | 87.40 | 87.45 | 87.44 | 87.06 | 86.97 | 86.82 |
| | WRN-50-2 | 88.27 | 88.26 | 88.10 | **88.30** | 87.99 | 88.25 | 87.97 | 87.96 | 87.75 | 87.58 |
| | WRN-50-4 | 88.64 | 89.09 | 89.00 | 89.08 | **89.12** | 88.95 | 88.94 | 88.98 | 88.46 | 88.39 |
| Pets | ResNet-18 | **91.94** | 91.81 | 90.79 | 91.59 | 91.09 | 90.46 | 89.49 | 87.96 | 84.83 | 82.41 |
| | ResNet-50 | 93.49 | **93.61** | 93.50 | 93.59 | 93.34 | 93.06 | 92.50 | 92.09 | 89.41 | 88.13 |
| | WRN-50-2 | 93.96 | 94.05 | 93.98 | **94.23** | 94.02 | 94.02 | 93.39 | 93.07 | 90.80 | 89.76 |
| | WRN-50-4 | 94.20 | **94.53** | 94.40 | 94.38 | 94.27 | 94.11 | 94.02 | 93.79 | 92.91 | 91.94 |
| SUN397 | ResNet-18 | **59.41** | 58.98 | 59.19 | 58.83 | 58.61 | 58.29 | 58.14 | 56.97 | 55.14 | 54.23 |
| | ResNet-50 | **62.24** | 62.12 | 61.93 | 61.89 | 61.50 | 61.64 | 61.28 | 60.66 | 59.27 | 58.40 |
| | WRN-50-2 | 63.02 | 63.28 | 63.16 | 63.18 | 62.90 | **63.36** | 62.53 | 62.23 | 61.16 | 60.47 |
| | WRN-50-4 | 63.72 | **64.89** | 64.81 | 64.71 | 64.74 | 64.53 | 64.49 | 64.74 | 62.86 | 62.14 |

## G.3 Unifying dataset scale

### G.3.1 Fixed-feature (cf. Fig. 7 & 15)

Table 10: **Fixed-feature** transfer on 32x32 downsampled datasets.

| | | Transfer Accuracy (%) | | | | | | | | | |
|---|---|---|---|---|---|---|---|---|---|---|---|
| | | Robustness parameter $\varepsilon$ | | | | | | | | | |
| Dataset | Model | 0.00 | 0.01 | 0.03 | 0.05 | 0.10 | 0.25 | 0.50 | 1.00 | 3.00 | 5.00 |
| Aircraft | ResNet-18 | 17.64 | 18.72 | 19.11 | 20.34 | 21.69 | 23.19 | 24.93 | 25.44 | **27.15** | 26.01 |
| | ResNet-50 | 15.87 | 17.04 | 17.82 | 18.48 | 20.19 | 22.44 | 24.12 | 25.89 | **28.59** | 28.35 |
| Birdsnap | ResNet-18 | 14.76 | 14.04 | 15.80 | 16.23 | 17.77 | 18.60 | 19.75 | **20.16** | 19.15 | 16.72 |
| | ResNet-50 | 13.85 | 14.12 | 14.67 | 15.42 | 16.94 | 19.67 | 21.74 | **23.08** | 22.98 | 20.70 |
| CIFAR-10 | ResNet-18 | 76.02 | 74.36 | 79.48 | 79.71 | 82.97 | 86.62 | 88.47 | 90.29 | **91.64** | 90.36 |
| | ResNet-50 | 79.63 | 82.18 | 82.15 | 83.88 | 85.41 | 89.35 | 91.13 | 92.89 | **94.81** | 94.23 |
| CIFAR-100 | ResNet-18 | 54.61 | 54.03 | 58.77 | 58.74 | 63.64 | 68.10 | 70.66 | 72.74 | **74.01** | 72.08 |
| | ResNet-50 | 58.01 | 60.17 | 60.87 | 63.24 | 65.73 | 71.32 | 74.19 | 77.17 | **79.50** | 78.27 |
| Caltech-101 | ResNet-18 | 52.88 | 54.20 | 62.56 | 60.43 | 65.31 | 69.39 | 69.08 | 72.11 | **73.02** | 70.04 |
| | ResNet-50 | 56.55 | 59.32 | 60.45 | 61.08 | 63.76 | 69.80 | 73.11 | 76.89 | **78.86** | 77.43 |
| Caltech-256 | ResNet-18 | 40.60 | 40.83 | 45.02 | 45.88 | 49.96 | 51.08 | 51.36 | **54.13** | 53.79 | 51.87 |
| | ResNet-50 | 42.73 | 45.11 | 45.65 | 47.52 | 49.61 | 53.63 | 56.12 | 58.93 | **59.79** | 58.67 |
| Cars | ResNet-18 | 13.88 | 14.18 | 16.14 | 16.95 | 19.61 | 20.20 | 20.33 | **21.70** | 20.89 | 18.75 |
| | ResNet-50 | 13.16 | 13.89 | 13.68 | 16.84 | 17.07 | 19.40 | 21.88 | 23.19 | **24.19** | 23.37 |
| DTD | ResNet-18 | 35.96 | 36.33 | **40.27** | 37.87 | 39.79 | 39.31 | 39.73 | 40.05 | 39.10 | 39.41 |
| | ResNet-50 | 41.28 | 40.37 | 41.06 | 42.13 | 41.22 | 43.56 | 44.10 | 43.78 | 43.83 | **44.26** |
| Flowers | ResNet-18 | 64.81 | 65.75 | 70.01 | 70.57 | 72.71 | 74.46 | 74.19 | **76.06** | 74.23 | 71.52 |
| | ResNet-50 | 66.65 | 68.49 | 68.24 | 71.03 | 73.12 | 75.83 | 76.52 | 77.23 | **78.31** | 75.71 |
| Food | ResNet-18 | 31.58 | 32.98 | 35.98 | 36.42 | 38.46 | 39.35 | 39.56 | **41.22** | 40.17 | 38.35 |
| | ResNet-50 | 36.46 | 36.82 | 36.37 | 39.85 | 40.91 | 43.08 | 44.88 | 46.16 | **46.45** | 44.44 |
| Pets | ResNet-18 | 48.74 | 46.98 | 56.87 | 56.25 | 61.92 | 62.45 | 63.39 | **66.20** | 62.23 | 57.15 |
| | ResNet-50 | 53.98 | 54.10 | 58.55 | 53.57 | 59.58 | 67.35 | 69.31 | **70.16** | 69.43 | 64.37 |
| SUN397 | ResNet-18 | 23.16 | 24.35 | 25.34 | 25.94 | 27.60 | 28.00 | 28.12 | 30.19 | **30.91** | 30.41 |
| | ResNet-50 | 23.62 | 25.60 | 24.64 | 27.30 | 27.56 | 29.24 | 31.36 | 32.37 | **33.90** | 33.58 |

## G.3.2 Full-network (cf. Fig. 16)

Table 11: **Full-network** transfer on 32x32 downsampled datasets.

| | | Transfer Accuracy (%) | | | | | | | | | |
|---|---|---|---|---|---|---|---|---|---|---|---|
| | | Robustness parameter $\varepsilon$ | | | | | | | | | |
| **Dataset** | **Model** | 0.00 | 0.01 | 0.03 | 0.05 | 0.10 | 0.25 | 0.50 | 1.00 | 3.00 | 5.00 |
| Aircraft | **ResNet-18** | 58.24 | 58.27 | 59.29 | 58.96 | 60.28 | 60.22 | 59.83 | 60.88 | **61.78** | 60.88 |
| | **ResNet-50** | 65.77 | 65.20 | 65.62 | 66.22 | 65.68 | 67.12 | 66.49 | 66.04 | **68.02** | 67.12 |
| Birdsnap | **ResNet-18** | 46.32 | 46.65 | 45.94 | 46.55 | 46.26 | 46.57 | 46.26 | **46.80** | 45.23 | 44.76 |
| | **ResNet-50** | 52.28 | 51.98 | 51.77 | 52.11 | 52.20 | 52.42 | **52.58** | 51.77 | 51.72 | 51.29 |
| CIFAR-10 | **ResNet-18** | 96.50 | 96.38 | 96.51 | 96.62 | 96.78 | 96.86 | 97.12 | 97.04 | **97.14** | 97.05 |
| | **ResNet-50** | 97.30 | 97.32 | 97.54 | 97.56 | 97.62 | 97.79 | 97.98 | 98.10 | **98.27** | 98.16 |
| CIFAR-100 | **ResNet-18** | 82.36 | 82.57 | 82.89 | 82.92 | 83.31 | 83.90 | 84.30 | **84.41** | 83.77 | 83.47 |
| | **ResNet-50** | 85.15 | 85.37 | 85.64 | 85.68 | 85.92 | 86.45 | 86.81 | 87.32 | **87.45** | 86.60 |
| Caltech-101 | **ResNet-18** | 79.33 | 78.64 | 78.95 | 79.94 | 79.70 | 81.13 | 81.55 | **83.13** | 82.30 | 79.80 |
| | **ResNet-50** | 82.18 | 83.05 | 84.50 | 84.72 | 84.74 | 85.62 | 86.12 | **86.61** | 85.88 | 85.20 |
| Caltech-256 | **ResNet-18** | 63.32 | 64.45 | 64.02 | 64.55 | 65.18 | 66.00 | **66.52** | 65.41 | 64.35 | 63.03 |
| | **ResNet-50** | 68.02 | 68.09 | 68.63 | 69.42 | 68.96 | 70.10 | 70.60 | **70.66** | 69.90 | 68.94 |
| Cars | **ResNet-18** | 68.83 | 68.55 | 68.62 | 68.98 | 69.53 | 69.28 | **69.68** | 69.27 | 67.99 | 67.42 |
| | **ResNet-50** | 74.84 | 74.95 | 74.13 | 75.23 | 74.61 | 75.29 | **75.92** | 75.51 | 75.19 | 74.65 |
| DTD | **ResNet-18** | 49.57 | 48.40 | 50.43 | 48.88 | 49.20 | 50.27 | 50.00 | **50.74** | 50.32 | **50.74** |
| | **ResNet-50** | 50.69 | 52.50 | 51.01 | 51.60 | 51.65 | 52.66 | 54.15 | 52.71 | 54.26 | **55.53** |
| Flowers | **ResNet-18** | 85.96 | 86.05 | 86.02 | 86.03 | 86.40 | 86.25 | **86.41** | 86.03 | 85.33 | 84.60 |
| | **ResNet-50** | 88.75 | 88.30 | 88.57 | 88.27 | **88.81** | 88.69 | 88.70 | 88.37 | 88.67 | 87.83 |
| Food | **ResNet-18** | 71.77 | 71.83 | 71.73 | 71.64 | 71.60 | 71.64 | **72.10** | 71.63 | 71.78 | 71.37 |
| | **ResNet-50** | 75.83 | 75.19 | 75.52 | 75.51 | 75.50 | 75.37 | **76.11** | 75.91 | 75.76 | 75.61 |
| Pets | **ResNet-18** | 76.32 | 77.35 | 77.71 | 78.05 | 78.63 | 78.70 | **78.75** | 77.82 | 75.72 | 72.21 |
| | **ResNet-50** | 82.34 | 81.95 | 82.64 | 82.24 | 82.52 | 83.59 | 83.57 | **83.72** | 81.87 | 79.33 |
| SUN397 | **ResNet-18** | 42.81 | 42.65 | 43.40 | 43.35 | 44.01 | 44.20 | 44.51 | **44.61** | 44.31 | 43.54 |
| | **ResNet-50** | 44.64 | 44.95 | 44.73 | 45.09 | 45.44 | 45.93 | 46.74 | 47.24 | **47.47** | 47.15 |

## Footnotes

[2]These models can be found here https://pytorch.org/docs/stable/torchvision/models.html

[3]WRN-50-2 and WRN-50-4 refer to Wide-ResNet-50, twice and four times as wide, respectively.

[4]For all of our experiments, we do not freeze the batch statistics, only its weights.

[5]See: https://github.com/facebookresearch/detectron2/blob/master/MODEL_ZOO.md For all COCO tasks we used "R50-FPN" configurations (1x and 3x, described further in this section), and for VOC we used the "R50-C4" configuration.

[6]https://pytorch.org/docs/stable/torchvision/index.html

[7]https://github.com/KaimingHe/deep-residual-networks

[8]See for both previous note and model differences: https://github.com/facebookresearch/detectron2/blob/master/tools/convert-torchvision-to-d2.py

[9]Baselines found here: https://github.com/facebookresearch/detectron2/blob/master/MODEL_ZOO.md