[Reviews · NeurIPS 2020]

Review 1

Summary and Contributions: The paper studies transfer learning of adversarially robust models. The main finding is that the adversarially robust models, while being less accurate than their standard counterparts, can match or exceed transfer performance of their counterparts. The study is well supported by an extensive empirical evaluation.

Strengths: 1. Better understanding of transfer learning is of high importance for both using deep networks in practice and advancing our understating of deep networks which makes this work very relevant to the community. 2. The paper studies an interesting aspect of transfer learning (the effect of adversarial robustness) and presents interesting findings. 3. The study is well-supported by a thorough empirical evaluation. Models are trained on ImageNet and evaluated on a range of downstream classification datasets and tasks (detection, instance segmentation). The authors also perform a range of ablation studies (S4). 4. The paper is well-written, easy to follow, and the authors are careful when interpreting the results (e.g. L172, L204).

Weaknesses: 1. Are the COCO accuracies for each model averaged across multiples re-runs / seeds? If not, it would be good to do so since the COCO results can vary typically up to ~0.3 AP across re-runs. If yes, it would be good to clarify this. 2. It would be good to verify the main findings (e.g. Table 1) for different model types (e.g. ResNeXt, EfficientNet). Models with SE or non-local operators may be particularly interesting. Although there is some data on using different models at fixed robustness levels in Table 2 and Appendix G.2 it is not easy to conclude from that alone. 3. Width experiments from S4.2: I think it’s hard to draw general conclusions about the impact of width from the presented experiments for two main reasons: (a) there are only 4 data points and (b) varying network complexity may be a confounding factor. It would be better to perform this experiments using more data points and while controlling for complexity (same complexity, varying width), similar to population-level studies from [1]. I note and appreciate that the authors appropriately scope their observations (L173). 4. Dataset granularity hypothesis S4.3: While the image resolution experiments can serve as a crude proxy I think it’s hard to draw general conclusions from that. I think that it may be more convincing to perform experiments on a dataset for fine-grained recognition (e.g. iNaturalist). Again, the authors scope their observations appropriately (L204). References: [1] Radosavovic et al, On Network Design Spaces for Visual Recognition, ICCV 2019 Updated review: I thank the authors for the response. I keep my original score. I would encourage the authors to clarify / address (1) from above in the revised version of the paper.

Correctness: The paper performs an extensive empirical study. The authors consider realistic models, datasets, and tasks (e.g. ImageNet, COCO). The experiments are well-designed and carefully controlled.

Clarity: The paper Is well-written and easy to follow.

Relation to Prior Work: The prior work is discussed clearly throughout text, in Section 5, and in Appendix E.

Reproducibility: Yes

Additional Feedback: Minor comments: - L19: Consider citing [1, 2] in addition to [Don+14, Sha+14] - L90: undefined ref - L91: considers -> consider - Figure 4: semantic segmentation -> instance segmentation Updated review: I thank the authors for the response. - Figure 9: semantic segmentation -> instance segmentation - It may be clearer to use model names instead of A-F in tables References: [1] Girshick et al, Rich feature hierarchies for accurate object detection and semantic segmentation, CVPR 2014 [2] Oquab et al, Learning and transferring mid-level image representations using convolutional neural networks, ECCV 2014


Review 2

Summary and Contributions: This paper study the transfer learning performance for the CNN models trained using adversarial robust objective. The experimental results have shown improvement over CNN models trained with standard objective on a a varieties of image classification datasets, using fixed feature extractor or end-to-end finetuning. It also improve the object detection results on MS-COCO and VOC dataset.

Strengths: The improvement is relatively consistent across multiple datasets on image classification. This work can inspire other work in the transfer learning area.

Weaknesses: 1. Lack of novelty. The transfer learning performance of adversarial robustness training has been studied in prior work [Mad 18] and "Why Do Adversarial Attacks Transfer?" 2. The improvement is marginal, with many datasets with less than 0.5%. The improvement on COCO object detection is 0.3~0.5, which is close to the error margin.

Correctness: Yes. The experimental results can support the conclusion of this paper, adversarial robustness is helpful for transfer learning.

Clarity: This paper lacks of technical details. There are few notations for the only equation of this paper. The readers are not supposed to read previous paper to understand the method. Lack of experimental details during the pretraining and transfer learning, the reader is not able to fully reproduce the experiments base on the paper.

Relation to Prior Work: As discussed, there are minor novelties comparing to prior work.

Reproducibility: Yes

Additional Feedback:


Review 3

Summary and Contributions: This paper provides a thorough study of the robust models for transfer learning tasks. The authors clearly point out the conventional belief on transfer learning, especially for pretrained model, provide the motivation (which is to investigate whether the accuracy of pre-trained tasks is the only way to improve performance), and hypothesize that the robustness of models is a crucial factor for transfer learning tasks. To verify their hypothesis, the authors provide an extensive empirical study on twelve datasets for the image classification task, two datasets on object detection tasks, and instance segmentation tasks. In addition, they also provide a convincing explanation and discussion on pre-text accuracy for transfer learning performance, model width, optimal robustness level, and comparison to texture robustness.

Strengths: + Clear motivation and interesting hypothesis: the authors point out the previous belief on the pretrained model, in which the accuracy is relevant to the transfer learning performance. And, this leads to an interesting hypothesis, whether robustness of model determines the transfer learning performance. + Convincing results by extensive experiments: to verify the previous hypothesis, the authors provide an extensive empirical study on three different tasks and more than ten datasets. The experimental results demonstrate the consistent observation that a robust model can perform better in the transfer learning tasks. + Insightful discussion: the authors also provide several interesting discussions on the accuracy of source task (i.e., imagenet pre-training) for transfer learning performance, model width, optimal robustness level, and comparison to texture robustness. + Impactful for future research: this paper further provides a new direction for future research: why does a robust model perform better in transfer learning tasks.

Weaknesses: - I do not have any complaint about this paper.

Correctness: Given the experiment details in paper and the code provided by authors, there is no error found.

Clarity: This paper is well-written and provides a clear motivation, hypothesis, empirical studies and discussions. All sections are easy to read.

Relation to Prior Work: The authors provides a clear explanatio on prior works, and also point out a alternative hypothesis different from previous belief. And they provide extensive experiments to convince readers.

Reproducibility: Yes

Additional Feedback:


Review 4

Summary and Contributions: This paper provides a detailed empirical study to connect adversarially robust ImageNet classification models with their transferred accuracy in several downstream visual tasks. The extensive experiments answer a few interesting questions and provide useful insights to the community.

Strengths: 1. Although this paper does not propose new theoretical insights, it answers a few interesting questions of transfer learning in computer vision with extensive experiments. 2. It is the first to study the connection between adversarially robust models and transferred downstream task accuracy. 3. This paper provides a good explanation for two contradictory arguments: “More accuracy model transfers better” and “Adversarially robust model transfers better (while adversarially robust model usually has lower accuracy in ImageNet classification)”

Weaknesses: 1. Clarity - I find Figure 5 a bit confusing, maybe the authors can further elaborate it 2. Experiments - For the comparison with texture robust models, I wonder how the authors train these models. Are they trained with Stylized ImageNet only? Since there are two different options in the original paper [Gei+19]: train with Stylized ImageNet + ImageNet; train with Stylized ImageNet + ImageNet and then fine-tune on ImageNet. 3. Open questions Seems that the robustness level is very important, I wonder if the authors have any insights on how to select the robustness level for a new dataset.

Correctness: The authors make a few assumptions and validate them well with extensive experiments.

Clarity: Yes, the overall logical flow of the whole paper is clear. I especially like that the authors start preliminary studies on fixed feature transfer and then naturally extend to full-network fine-tuning.

Relation to Prior Work: Yes. Previous papers study different inductive bias to improve the performance on the downstream tasks, adversarial robustness can be regarded as another inductive bias.

Reproducibility: Yes

Additional Feedback: Update: I have read the comments from other reviewers and the rebuttal, I think the authors well address most of the raised issues. Thus, I keep my original rating and recommend acceptance.

[Author Response · NeurIPS 2020]

We thank the reviewers for their comments. We address individual concerns below.

**Reviewer 1:** *It would be good to verify the main findings (e.g. Table 1) for different model types.* We agree that
this would be interesting, the main reason we did not include a full analysis of more architectures is computational
constraints, since each architecture requires its own grid search over the various hyperparameters outlined in the paper.
We believe that the results in G.2 are enough to demonstrate that the increased performance is a robust phenomenon,
and we hope to inspire future works that test transfer learning performance across more architectures and datasets.

*Width experiments from S4.2.* We agree that running with even wider architectures would lead to a better understanding
of the trend. Unfortunately, the number of distinct architectures we can train is again bounded by computational
constraints. We would like to note that, particularly in the fixed-feature setting, the trend of "eps=0 network increases
then plateaus/decreases" holds robustly across datasets, which we believe gives some indication of the generality of
the phenomenon. Also, ResNet-50, WRN-50x2, and WRN-50x4 are all ResNet-50 models with varying width (only
ResNet-18 differs in architecture).

*Dataset granularity hypothesis S4.3.* We thank the reviewer for the suggestion. Though we agree that resolution is a
coarse proxy, we wanted to focus S4.3 on a quantitative notion of granularity. It would be very interesting future work
to test the same relationship with respect to other more complex quantitative notions of dataset granularity.

**Reviewer 2:** *Lack of novelty.* We believe the reviewer is conflating "transfer learning" (wherein one uses a pre-trained
classifier on one dataset to perform better on another dataset) with "adversarial transfer" (the phenomenon where
adversarial attacks that fool one architecture tend to also fool another architecture). The two fields are entirely
unrelated—our work is on the former, whereas [Mad+18] and others discuss the latter.

*The improvement is marginal.* While the improvement is sometimes small, note that (a) robust models consistently
outperform standard models, which adds significance to the result, (b) on many datasets the improvement given by
robust models is outside of error bars, and (c) that robust models have much worse accuracy than standard models,
making even modest improvements somewhat surprising.

*Lack of clarity #1 (technical details).* We are not sure what the reviewer means by this comment. The equation given is
fairly standard, and in Appendix F we give a detailed primer on adversarial robustness which introduces each symbol in
the first equation explicitly, and also provides other background technical knowledge.

*Lack of clarity #2 and lack of reproducibility (experimental details).* We are again confused by the reviewer's comment
here, since Appendix A in the supplementary materials provides all of the details necessary to reproduce the experiments.
Furthermore, we provide a full code release (the link is in the paper) with all of our pre-trained ImageNet models and
easy-to-run code for reproducing any of the numbers reported in our paper.

*Related work.* We hope that the reviewer's concern is alleviated by the clarification above (i.e., the difference between
"transfer learning" and "adversarial transfer.") In both Section 5 and Appendix E we outline and discuss all of the related
work of which we are aware.

**Reviewer 5:** Thank you for your comments!

**Reviewer 6:** *Clarity: Figure 5 a bit confusing.* Thanks for pointing this out. Figure 5 summarizes the results of our
fixed-feature transfer learning experiment on various datasets and architectures (dataset names are given above each
plot and architecture name below each plot group). Each data point corresponds to an ImageNet model pre-trained with
a given robustness level denoted by one of the markers (the legend at the top relates the marker to the robustness level).
The x coordinate is the clean accuracy of this model, and the y coordinate is corresponding transfer accuracy on the
relevant dataset.

Note that due to a formatting error the y axis legend (which should read "Transfer Accuracy") got cut off, we have fixed
this in the updated manuscript. We will also make sure to clarify the figure in the updated version of the paper.

*Experiments: comparison with texture robust models.* We train only on Stylized ImageNet.

*Open questions: insights on how to select the robustness level for a new dataset.* This is a great question. From our
analysis in section 4.3, it seems that the robustness level is correlated with the scale of the datasets; as the scale of the
dataset increases, the "best" corresponding robustness level decreases. One might be able to fit a function mapping
dataset scale to robustness level using the results of our experiments on various datasets (the have various scales)

We believe more analysis and experiments are required before reaching conclusions on how to select the best robustness
level.

[Meta-Review · NeurIPS 2020]

This paper is a clear accept, with a consensus amongst reviewers. It has a clear motivation and proposes an interesting hypothesis, which in turn is well-supported by a thorough empirical evaluation. The paper is well-written, easy to follow, and the authors are careful when interpreting the results.